# Salivary MicroRNA Signature for Diagnosis of Endometriosis

**DOI:** 10.3390/jcm11030612

**Published:** 2022-01-26

**Authors:** Sofiane Bendifallah, Stéphane Suisse, Anne Puchar, Léa Delbos, Mathieu Poilblanc, Philippe Descamps, Francois Golfier, Ludmila Jornea, Delphine Bouteiller, Cyril Touboul, Yohann Dabi, Emile Daraï

**Affiliations:** 1Department of Obstetrics and Reproductive Medicine, Hôpital Tenon, 4 Rue de la Chine, 75020 Paris, France; anne.puchar@aphp.fr (A.P.); cyril.touboul@gmail.com (C.T.); yohann.dabi@gmail.com (Y.D.); emile.darai@aphp.fr (E.D.); 2Clinical Research Group (GRC) Paris 6, Centre Expert Endométriose (C3E), Sorbonne University (GRC6 C3E SU), 4 Rue de la Chine, 75020 Paris, France; 3Ziwig Health, 19 Rue Reboud, 69003 Lyon, France; stephane@ziwig.com; 4Department of Obstetrics and Reproductive Medicine, Centre Hospitalier Universitaire, 49000 Angers, France; lea.delbos@chu-angers.fr (L.D.); phdescamps@chu-angers.fr (P.D.); 5Endometriosis Expert Center, Pays de la Loire, 49000 Angers, France; 6Department of Obstetrics and Reproductive Medicine, Lyon South University Hospital, Lyon Civil Hospices, 69008 Lyon, France; mathieupoilblanc@gmail.com (M.P.); francois.golfier@chu-lyon.fr (F.G.); 7Endometriosis Expert Center, Steering Committee of the EndAURA Network, 75020 Paris, France; 8Paris Brain Institute—Institut du Cerveau—ICM, Inserm U1127, CNRS UMR 7225, AP-HP—Hôpital Pitié-Salpêtrière, Sorbonne University, 75020 Paris, France; ludmila.jornea@icm-institute.org; 9Genotyping and Sequencing Core Facility, iGenSeq, Institut du Cerveau et de la Moelle Epinière, Institut du Cerveau, Hôpital Pitié-Salpêtrière, 47-83 Boulevard de l’Hôpital, 75013 Paris, France; delphine.bouteiller@icm-institute.org

**Keywords:** endometriosis, saliva, diagnostic, signature, miRNA

## Abstract

Background: Endometriosis diagnosis constitutes a considerable economic burden for the healthcare system with diagnostic tools often inconclusive with insufficient accuracy. We sought to analyze the human miRNAome to define a saliva-based diagnostic miRNA signature for endometriosis. Methods: We performed a prospective ENDO-miRNA study involving 200 saliva samples obtained from 200 women with chronic pelvic pain suggestive of endometriosis collected between January and June 2021. The study consisted of two parts: (i) identification of a biomarker based on genome-wide miRNA expression profiling by small RNA sequencing using next-generation sequencing (NGS) and (ii) development of a saliva-based miRNA diagnostic signature according to expression and accuracy profiling using a Random Forest algorithm. Results: Among the 200 patients, 76.5% (n = 153) were diagnosed with endometriosis and 23.5% (n = 47) without (controls). Small RNA-seq of 200 saliva samples yielded ~4642 M raw sequencing reads (from ~13.7 M to ~39.3 M reads/sample). Quantification of the filtered reads and identification of known miRNAs yielded ~190 M sequences that were mapped to 2561 known miRNAs. Of the 2561 known miRNAs, the feature selection with Random Forest algorithm generated after internally cross validation a saliva signature of endometriosis composed of 109 miRNAs. The respective sensitivity, specificity, and AUC for the diagnostic miRNA signature were 96.7%, 100%, and 98.3%. Conclusions: The ENDO-miRNA study is the first prospective study to report a saliva-based diagnostic miRNA signature for endometriosis. This could contribute to improving early diagnosis by means of a non-invasive tool easily available in any healthcare system.

## 1. Introduction

Endometriosis, defined by the presence of endometrium-like tissue outside the uterus, affects 2–10% of the female population, i.e., around 190 million women worldwide. It is a heterogenous disease with a poorly understood natural history and as such poses many challenges [1,2]. The first is timely diagnosis mainly because the symptoms of endometriosis are non-specific, and clinical examination is often either negative or results in a wrong diagnosis [3,4]. The second challenge is that complementary explorations, especially biomarkers [5,6] and imaging examinations [5,6,7,8], are often inconclusive and fail to diagnose early-stage endometriosis with sufficient accuracy or are of limited relevance for the severe forms. Consequently, therapeutic and follow-up strategies are compromised, and there is a high rate of conventional treatment failure [2,4]. Finally, endometriosis constitutes a considerable economic burden for the healthcare system linked not only to direct costs but also to indirect costs from school and work absenteeism. Overall, the annual cost of endometriosis was estimated at around 10,000 euros per patient in 2012, which is equivalent to that of diabetes in France in 2017 [9,10,11,12,13].

During the last decade, new diagnostic tools have been investigated to detect this debilitating disorder as early as possible [5,14,15,16,17,18]. Among these, microRNA (miRNA) analysis is emerging as a promising option supported by a growing body of evidence from studies in cancer and degenerative disorders [19,20,21,22,23]. Human miRNAs are single stranded, highly conserved, non-coding RNAs composed of 21–25 nucleotides. Partial binding to their complementary messenger RNA (mRNA) can regulate gene degradation and translation. It is estimated that about 60% of genes are regulated by miRNAs [24]. From a biological point of view, miRNAs are mainly transcribed from genes in intronic regions of coding or non-coding transcripts [24,25,26]. miRNAs are transcribed in the nucleus under hundreds of duplex nucleotide-long primary miRNAs (pri-miRNA) subsequently cleaved to generate precursor miRNAs (pre-miRNA). These pre-miRNAs are then transported from the nucleus to the cytoplasm, where the duplexes are cleaved to form mature miRNAs, which are incorporated into RNA silencing complexes (RISC) that regulate post-translational modifications by binding to the target mRNA [24,25,26]. Finally, the miRNAs are released from the cells into the circulation using various carriers, such as Argonaute, nucleophosmin 1, high-density lipoproteins, or extracellular vesicles (exosomes), which confer remarkable stability against endogenous RNAses. The miRNAs can then be detected in human fluids [24,25,26].

In the specific setting of endometriosis, several authors have evaluated the relevance of a blood-based miRNA signature, but the results are discordant because of methodological and control group issues [19,20,22,23,27,28]. Indeed, in previous studies, the control groups was composed of asymptomatic patients and/or patients undergoing a tubal ligation and/or pelvic inflammatory disease and/or gynecologic disorders but without knowledge of symptoms [19,20,22,23].

Similarly, saliva miRNAome analysis has been investigated by several teams exploring biomarkers for numerous benign and malignant disorders but never in the context of endometriosis [29,30,31,32].

Therefore, the aim of the prospective ENDO-miRNA study was to analyze the human miRNome to differentiate patients with and without endometriosis, to define a saliva-based diagnostic miRNA signature for endometriosis with an internal cross validation.

## 2. Material and Methods

### 2.1. Ethics Statement

The data and saliva used for analysis were collected from the prospective ENDO-miRNA study (ClinicalTrials.gov Identifier: NCT04728152) under the Research Protocol ID RCB: 2020-A03297-32. Informed consent was obtained from all the participants. The study and data analysis followed the Standards for Reporting of Diagnostic Accuracy (STARD) reporting guidelines [33] (Appendix A). The study consisted of two parts: (i) identification of a biomarker based on genome-wide miRNA expression profiling by small RNA sequencing using next-generation sequencing (NGS) and (ii) development of a saliva-based miRNA diagnostic signature according to expression and accuracy profiling using an ML algorithm [17,21,34,35,36,37,38,39,40,41,42].

### 2.2. Study Population

The prospective ENDO-miRNA study included 200 saliva samples obtained from women with chronic pelvic pain suggestive of endometriosis. All the saliva samples were collected between January and June 2021. All the patients underwent either a laparoscopic procedure (therapeutic or diagnostic laparoscopy) and/or MRI imaging [5,6,7,8]. The laparoscopic procedures were systematically videoed and then analyzed by two operators (C.T., Y.D.), blinded to the symptoms and imaging findings, to confirm the presence or absence of endometriosis. For the patients who underwent laparoscopy, diagnosis was confirmed by histology. For the patients diagnosed with endometriosis without laparoscopic evaluation, all had MRI with features of deep endometriosis with colorectal involvement and/or endometriomas confirmed by two expert radiologists. The miRNAs were analyzed blinded to the surgical and imaging findings. Following exploration by laparoscopy or MRI, the women were classified into two groups: an endometriosis group and a control group of women with various benign pathologies other than endometriosis or with symptoms suggestive of endometriosis but without clinical or MRI features and no endometriosis lesions found during laparoscopic inspection (complex patient). The study flow chart is reported in Figure 1. The patients with endometriosis were stratified according to the revised American Society of Reproductive Medicine (rASRM) classification [43].

### 2.3. Saliva Sample Collection

The saliva samples (2 mL) were collected in an all-in-one system including a nucleic acid stabilizing solution for the collection, stabilization, and transportation (RE-100, DNA Genotek Inc., 2 Beaverbrook Road, Ottawa, ON, Canada) using an at-home kit (https://www.dnagenotek.com/us/products/collection-infectious-disease/OME-505.html, accessed on 1 December 2021). All the samples were stored at room temperature prior to shipping.

### 2.4. RNA Sample Extraction, Preparation and Quality Control

The RNA was isolated from each saliva sample using the miRNeasy Kit (Qiagen, Inc., Germantown, MD, USA) according to the manufacturer’s instructions [29,31,44,45]. RNA quality was assessed using the Agilent Technologies Tapestation 2200. RNA-sequencing libraries were prepared using the QIAseq miRNA Library Kit (Qiagen) according to the manufacturer’s instructions. Samples were indexed in batches of 96, with a targeted sequencing depth of 17 million reads per sample. Sequencing was performed using 100 base single-end reads, using an Novaseq6000 sequencer (Illumina, San Diego, CA, USA) [46,47].

## 3. Bioinformatics

### 3.1. Raw Data Preprocessing (Raw, Filtered, Aligned Reads) and Quality

Sequencing reads were processed using the data processing pipeline. FastQ files were trimmed to remove adapter sequences using Cutadapt version v.1.18 and were aligned using Bowtie version 1.1.1 with the following transcriptome databases: the human reference genome available from NCBI (https://www.ncbi.nlm.nih.gov/genome/guide/human/, accessed on 1 December 2021) and miRBase21 (miRNAs) using the MirDeep2 v0.1.0 package. The raw sequencing data quality was assessed using FastQC software v0.11.7 [40,44,46,48,49,50].

### 3.2. Differential Expression Analysis of miRNAs

miRNA expression was quantified by miRDeep2 v0.1.0 [51]. Differential expression tests were then conducted in DESeq2 for miRNAs with read counts in ≥1 of the samples. DESeq2 V1.20 integrates methodological advances with several novel features to facilitate a more quantitative analysis of comparative RNA-seq data using shrinkage estimators for dispersion and fold change [52]. The resulting matrix was filtered for expressed miRNAs [53]. The miRNAs were considered as differentially expressed if the absolute value of log2 fold change was >1.5 (upregulated) and <0.5 (downregulated). The *p*-value adjusted for multiple testing was <0.05 [52]. The Appendix A summarized and adapted the miRNA-nome Sequencing Analysis Pipeline used from the methods of Potla et al. [44].

## 4. Statistical Analysis

### 4.1. Development and Validation of the Diagnostic Model

Random Forest (RF) was considered to design the saliva signature [34,35,36,37,54,55,56]. Random Forest (RF) classifier is an ensemble method that trains several Decision Tree (DT) in parallel with bootstrapping followed by aggregation, jointly referred to as bagging. Bootstrapping indicates that several individual DTs are trained in parallel on various subsets of a training dataset using different subsets of available features. Bootstrapping ensures that each individual DT in the RF is unique, which reduces the overall variance of the RF classifier. For the final decision, RF classifier aggregates the decisions of individual DTs and consequently exhibits good generalization [54]. F1-score, sensitivity, specificity, and the ROC AUC were calculated to assess and compare the diagnostic performance of the diagnostic signature [57,58].

### 4.2. Validation of the Signature Accuracy

The accuracy and reproducibility of the signature were tested on 10 data sets randomly [41,59,60] composed of the same proportion of control and endometriosis patients. Each data set was randomly generated to conserve the initial ratio of endometriosis and control patient’s profile. Analysis was performed using Python (Python Software Foundation) with XGBoost 1.3.3, scikit-learn 0.19.1, and scipy 1.1 packages.

### 4.3. Other Statistical Analyses

Statistical analysis was based on the chi-square test as appropriate for categorical variables. Values of *p* < 0.05 were considered to denote significant differences. Data were managed with an Excel database (Microsoft, Redmond, WA, USA) and analyzed using R 2.15 software, available online (http://cran.r-project.org/, accessed on 1 December 2021).

## 5. Results

### 5.1. Description of the ENDO-miRNA Cohort

The clinical characteristics of the patients in the endometriosis and control groups are presented in Table 1. Among the 200 patients, 76.5% (n = 153) were diagnosed with endometriosis and 23.5% (n = 47) without (controls). In the control group, 51% (24) of the women had no abnormality and were defined as discordant or complex patients (Table 1). There were no significant differences in terms of age or BMI between the groups. The mean (±SD) time from symptom onset to diagnosis for endometriosis patients was 14.8 years (±17.88). In both groups, the patients had pain symptoms suggestive of endometriosis. Comparatively, for patients with and without endometriosis using Visual Analogical Scale (VAS), the dysmenorrhea/of dysmenorrhea (mean ± SD) were 6 ± 3.4 versus 5 ± 3.2, *p* < 0.001; dyspareunia was 5.28 ± 3.95 verus 4.95 ± 3.52, *p* < 0.001; and urinary pain during menstruation (mean ± SD) were 4.35 ±3.36 versus 2.84 ±2.76, *p* < 0.001. For the endometriosis patients, 52% (80) had rASRM stage I–II, and 48% (73) had stage III–IV.

### 5.2. Global Overview of miRNA Transcriptome

Small RNA-seq of 200 saliva samples yielded ~4.642 M raw sequencing reads (from ~13.7 M to ~39.3 M reads/sample). Pre-filtering and filtering steps retained 70% (~3.205 M) of initial raw reads. The majority of the filtered reads were of short read length. Quantification of the filtered reads and identification of known miRNAs yielded ~190 M sequences that were mapped to 2561 known miRNAs from miRBase v21. The number of expressed miRNAs ranged from 1250 (outlier) to 2561 per sample. The distribution of expressed miRNAs in the 200 saliva samples and the overall composition of processed reads is shown in Figure 2.

### 5.3. Feature Selection of miRNAs Relevant for a Diagnosis of Endometriosis

The expression and accuracy of the miRNA profiles were used to identify miRNAs related to endometriosis. Out of the 2561 known miRNAs, the feature selection method generated a subset of 109 miRNAs. The respective correlation and accuracy to diagnose endometriosis according to the F1-score, sensitivity, specificity, and AUC ranged from 18.9–87.7%, 11.6–99.4%, 8.5–97.9%, and 36.9–69.2%, respectively. Among the 109 miRNAs selected, 79% (n = 86) and 21% (n = 23) had an AUC value of <60 and ≥60% for correlation and accuracy, respectively; 83% (n = 91) and 17% (n = 18) had an F1-score ranging between 0–79%, and ≥80%, respectively; 83% (n = 91) and 17% (n = 18) had a sensitivity ranging between 0–79%, and ≥80%, respectively; and 83% (n = 91) and 17% (n = 18) had a specificity ranging between 0–79%, and ≥80%, respectively. Finally, 0% (n = 0) were identified as being downregulated and 6.4% (n = 7) as being upregulated. Appendix A summarizes the diagnostic accuracy of the 109 miRNAs selected in the signature.

### 5.4. Saliva-Based Diagnostic Signature for Endometriosis

The overall performance of the diagnostic signature composed of 109 mi RNAs (Random Forest model) against the 10 randomized datasets is reported in Table 2. The sensitivity, specificity, and AUC ranges from 80% to 96.8%, 80% to 100%, and 79.9% to 98.4%, respectively. The signature, after internal cross validation on 10 different data sets, obtained its higher accuracy with a respective sensitivity, specificity, and AUC of 96.7%, 100% and 98.3% (Table 2).

### 5.5. Relation between Pathophysiology of Endometriosis and miRNA Expression

Among the 109 miRNAs composing the endometriosis diagnostic signature, 77% (84) have been reported to be associated with pathophysiologic pathways for benign and malignant disorders (Appendix A). Only miR-34c-5p and miR-19b-1-5p have previously been reported in the field of endometriosis. Among the 109 mi RNA of the signature, 29 (27%) are associated with the main signaling pathways of endometriosis: PI3K/Akt, PTEN, Wnt/β-catenin, HIF1α/NF κB, and YAP/TAZ/EGFR (Appendix A).

## 6. Discussion

To the best of our knowledge, the ENDO-miRNA study is the first prospective study to report a saliva-based diagnostic miRNA signature for endometriosis. This could contribute to improving early diagnosis by means of a non-invasive tool easily available in any healthcare system. Its value lies in the combination of the intrinsic quality of miRNA to condense endometriosis phenotypes (and its heterogeneity) and the modeling power of AI. Its reproducibility is based on our bioinformatics approach of miRNA-sequencing analysis and a statistical approach designed to overcome the complexity and heterogeneity of endometriosis.

We hypothesized that a saliva-based miRNA signature for endometriosis would be a low-cost and scalable method allowing samples to be collected anywhere by anyone. The tool would thus be available for underprivileged populations unlike methods based on blood samples, which are blood-volume and temperature dependent, imposing complex logistics of collecting peripheral blood and transporting it to a laboratory for analysis.

Saliva is an increasingly attractive body fluid in the search for disease biomarkers [29,30,31,32,45,61]. miRNAs exhibit remarkable stability in severe conditions, such as extended storage [24,29,32,62]. Zheng et al. demonstrated that saliva is not affected by coagulation that could induce a release of miRNAs. This is especially crucial as many studies evaluating miRNA expression in endometriosis have been performed on serum [23,61]. Zhang et al. first developed a technique to stabilize the saliva and process RNA analysis. The average range of RNA content in 1 L of bodily fluids is as low as 0.01 mg in urine to as high as 11.2 mg in saliva [63]. Moreover, blood, leukocytes, and saliva have lower standard deviations in their RNA content (<50% on average) compared with serum and urine. The average concentration of the isolated RNAs from all bodily fluids can be classified into high (>20 ng/μL) for blood, leukocytes, saliva, and cell-free saliva and low (<10 ng/μL) for plasma, serum, urine, and cell-free urine. Finally, more than 90% of miRNAs in saliva are shared with blood, leukocytes, and plasma, which further supports its stability and reproducibility [63].

In the specific setting of endometriosis, and despite the various endometriosis phenotypes, we were able to build an endometriosis diagnostic signature. The most accurate signature in our model provides a sensitivity, specificity and AUC of 96.7%, 100% and 98.3%, respectively. These values testify to the high accuracy of the signature, supporting its clinical value, and raise the issue to revise the current diagnostic strategy for exploring patients with symptoms suggestive of endometriosis, based on a diagnostic laparoscopy.

Multiple diagnostic biomarkers have been suggested as screening and triage tests to diagnose endometriosis [14,15], but none of them are of sufficient accuracy, i.e., a sensitivity of 0.94 and a specificity of 0.79 [5,14,15]. In accordance with the 2011 Biomarkers Definitions Working Group, a biomarker is “a characteristic that can be objectively measured and evaluated as an indicator of normal biological or pathogenic processes, or as an indicator of pharmacologic response to therapeutic interventions” [64]. The present study was able to quantify and analyze the miRNAome for (i) discordant/complex patients (women with chronic pelvic pain suggestive of endometriosis and both negative clinical examination and imaging findings), (ii) women with early-stage (stage I–II rASRM) and advanced-stage (stage III–IV rASRM) endometriosis, and (iii) women with other gynecological disorders sharing symptoms of endometriosis. To subscribe to biomarker criteria, we hypothesized the relevance of the exhaustive evaluation of all miRNAs associated with endometriosis for 200 saliva samples to unearth the complexity of the disease and its heterogeneity. To our knowledge, this is the first exhaustive sequencing of the human saliva miRNAome in the specific context of endometriosis, and we show that 97.3% of all miRNAs are detectable in the saliva with a homogeneous stability of reads. Our analysis resulted in the selection of a set of 109 miRNAs robustly tested.

Several studies have reported aberrant expression of miRNAs in affected tissues or peripheral blood samples of patients with endometriosis [20,22,23,62,63,64]. Several miRNAs have been shown to be dysregulated during the pathogenic process of endometriosis [20,22,23]. Diagnostic power of several miRNAs has been assessed in endometriosis [19,20,22,23,65]. For example, Maged et al. have shown that serum miR-122 and miR-199a had a sensitivity of 95.6 and 100.0% and a specificity of 91.4 and 100%, respectively, for diagnosis of disease status in women. Thus, these miRNAs are putative serum biomarkers for endometriosis [66]. To date, Moustafa et al. [65] is the only team to have attempted to build a blood-based miRNA diagnostic signature for endometriosis composed of six miRNAs based on Random Forest analysis. In agreement with previous studies [40,41,42], it would appear illusory that so few miRNAs could reflect the diversity of a multifactorial disorder such as endometriosis, which involves multiple and poorly known signaling pathways. Therefore, we hypothesized the value of (i) analyzing a specific selection of miRNAs, which resulted in a selection of 109; (ii) reducing the number of features to improve the final accuracy; and finally, (iii) using Random Forest model with high accuracy, which supports the value of AI technology. Such an approach has previously been validated in a study showing that a 100-miRNA blood signature was sufficiently stable to provide almost the same classification accuracy across different types of cancers and platforms [40,41]. Previous studies have demonstrated that saliva miRNA expression analysis can differentiate Crohn’s disease from ulcerative colitis and are of value in head and neck, pancreatic-biliary tract, and oral cancers but failed to report a true diagnostic signature [29,31]. Nevertheless, Rapado-González et al. reported a salivary miRNA signature composed of 22 miRNAs in colorectal cancer patients vs. healthy individuals in the discovery phase [30]. Moreover, Cheng et al. demonstrated the relevance of miRNA saliva expression to diagnose Yang and Yin Deficiency with a panel of 81 and 96 miRNAs, respectively [67].

From a pathophysiologic point of view, after systematic review, among the 109 miRNAs of our endometriosis signature, only four miRNAs (miR-34c-5p, miR-19b-1-5p, miR-149-5p, and miR-378a-3p) have previously been reported in patients with endometriosis; 25 have not been reported previously, implying that further studies should be conducted to confirm their involvement in the pathophysiology of endometriosis; and the remaining 80 are known to be involved in various signaling pathways, such as PI3K/Akt, PTEN, Wnt/β-catenin, HIF1α/NF κB, and YAP/TAZ/EGFR, with potential therapeutic implications.

Some limitations of the present study deserve to be discussed. First, although 97.3% of the human miRNAome were detectable and analyzed in the saliva samples, we cannot rule out that the remaining 2.7% of miRNAs are not involved in endometriosis. Second, as for blood sampling, a potential bias could be linked to the use of hormonal treatment: some patients in both the endometriosis and control group had undergone prior hormonal treatment that might have affected miRNA expression. However, previous studies [19,65] have reported that no significant miRNA changes are observed either during the menstrual cycle or in response to sex steroid hormone therapies [19]. Finally, as previously mentioned for the blood-based miRNA signature, another potential bias could be related to the inclusion of patients with deep endometriosis and/or endometrioma without laparoscopic control in the endometriosis group. However, previous studies have demonstrated the high accuracy of MRI to diagnose endometrioma and deep endometriosis with colorectal involvement, reaching the criteria for a replacement and SnNout triage test [5,7]. Although, our prospective study is the largest available on miRNA and saliva (n = 200) [61], the sample size, especially concerning the control group (n = 47), and the internal cross validation warrants an external validation. Our signature exhibits higher accuracy in patients over 18 years old, and our population did not include adolescents. Therefore, it is not possible to extrapolate our results in this specific population, impending further studies.

## 7. Conclusions and Perspectives

Despite some limits of the current prospective study, our data support the use of a saliva-based diagnostic miRNA signature for endometriosis in the diagnosis care pathways after an external validation to confirm these results. Saliva sampling is a cheap and non-invasive process and can be repeated multiple times, thus potentially improving both the diagnostic and therapeutic management of patients through early identification and for all populations. Finally, beyond the context of endometriosis, our methodology could be used as a blueprint to investigate other pathologies, both benign and malignant.

## Figures and Tables

**Figure 1 jcm-11-00612-f001:**
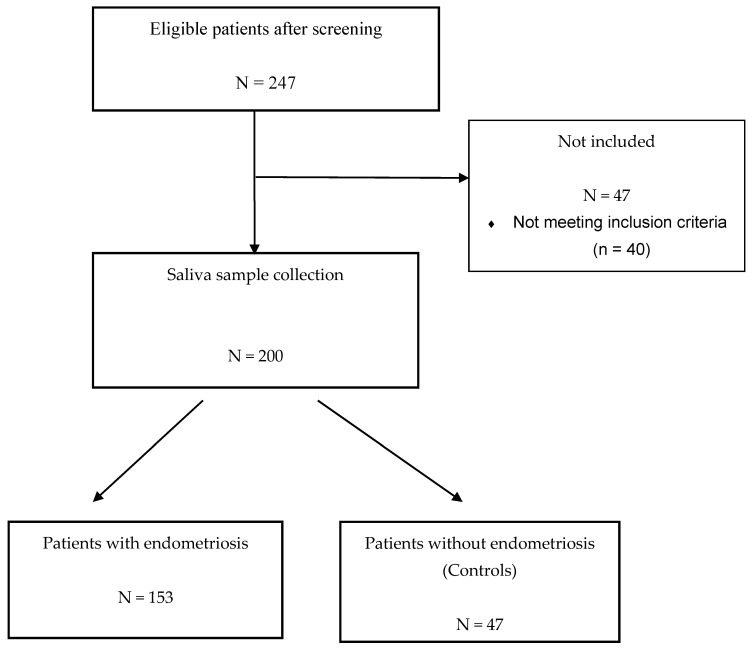
Flow chart of ENDO-miRNA study.

**Figure 2 jcm-11-00612-f002:**
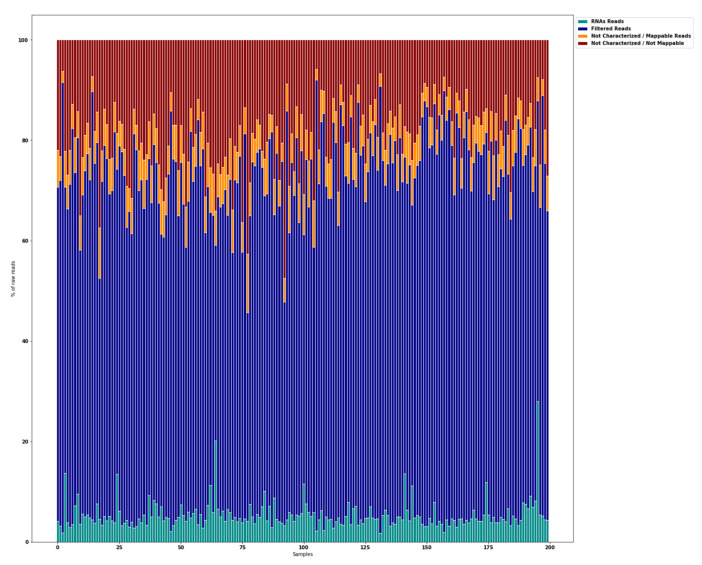
Overall composition of processed reads. RNA reads, miRNAs + piRNAs + rRNAs + tRNAs + mRNAs + others; Filtered Reads, reads with no adapters + reads with low quality bases + reads too short; Not Characterized/Mappable reads, mapped reads to GRCh38 that could not be characterized as a particular type; Not Characterized/Not Mappable reads, reads that could not be mapped.

**Table 1 jcm-11-00612-t001:** Demographic Characteristics of the population.

	Control Patients	Patients with Endometriosis	
	N (%)	N (%)	
	N = 47	N = 153	
Age (years) (mean ± SD)	30.92 (13.79)	31.17 (10.78)	0.1912
Age range			
- Less than 30 years	72% (34)	63% (96)	
- Over 30 years	28% (13)	37% (57)	0.294
BMI (body mass index) (mean ± SD)	24.84 (11.10)	24.36 (8.38)	0.525
Infertility			
- Yes	17% (8)	24% (36)	
- No	83% (39)	76% (117)	0.556
rASRM classification			-
- I–II	-	52% (80)	
- III–IV	-	48% (73)	
Control diagnoses			
- No abnormality	51% (24)	_	-
- Leiomyoma	2% (1)		
- Cystadenoma	11% (5)		
- Teratoma	23% (11)		
- Other gynecologic disorders	13% (6)		
Dysmenorrhea	100%	100%	
Abdominal pain outside menstruation			
- Yes	66% (21)	71% (89)	0.6905
Pain suggesting sciatica			
- Yes	31% (10)	56% (70)	0.0214
Lower back pain outside menstruation			
- Yes	62% (20)	81% (101)	0.0498
Right shoulder pain during menstruation			
- Yes	9% (3)	21% (26)	0.2184
Blood in the stools during menstruation			
- Yes	12% (4)	24% (30)	0.2425
Blood in urine during menstruation			
- Yes	25% (8)	17% (21)	0.4172
Diagnostic method			
- Surgery	47 (100)	83 (54.2)	
- Magnetic Resonance Imaging	-	70 (45.8)	-

**Table 2 jcm-11-00612-t002:** Random Forest accuracies for endometriosis diagnosis.

Random Forest			
Dataset	AUC	Sensitivity	Specificity
1	0.935	0.871	1
2	0.903	0.806	1
3	0.935	0.871	1
4	0.983	0.967	1
5	0.867	0.833	0.9
6	0.968	0.935	1
7	0.919	0.839	1
8	0.935	0.871	1
9	0.933	0.967	0.9
10	0.9	0.8	1

In yellow, the most accurate model.

## Data Availability

Data are available at sofiane.bendifallah@aphp.fr.

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
