# Peer review of "Salivary MicroRNA Signature for Diagnosis of Endometriosis"

_jcm, 2022, doi:10.3390/jcm11030612_

Round 1

Reviewer 1 Report

In this study the authors investigated the human miRNAome to define a saliva-based diagnostic miRNA signature for endometriosis in 200 women with chronic pelvic pain suggestive of endometriosis between January and June 2021. The aims of this study were the identification of a biomarker based on genome-wide miRNA expression profiling by small RNA sequencing using next generation sequencing (NGS), and the development of a saliva-based miRNA diagnostic signature according to expression and accuracy profiling using a Random Forest algorithm. Here, the first exhaustive sequencing of the human saliva miRNAome in the specific context of endometriosis has been reported resulting in the selection of a set of 109 miRNAs robustly tested.

  • Main points:
  • The authors may improve the introduction section by adding more previous studies reporting the miRNA previously reported in patients with endometriosis.
  •  In materials and methods section, and specifically in the paragraph 2.1 (“Ethics Statement”) at line 97, the authors could insert the full form for the abbreviation “ML” and could explain better this algorithm in the appropriate paragraph. In the paragraph 2.2 (“Study population”), the authors could better describe the collection of demographic characteristics of the patients included in the cohort and the methods used to measure them. Specifically, the authors could explain the methods used to measure some parameters such as the pains described. The authors could also insert the reference where the parameters used have been previously described. Moreover, in the flow chart of the study are indicated the eligible patients after screening (247). Subsequently, 47 patients have been excluded by the study and the authors could better explain the not meeting inclusion criteria used.
  • In the Results section, and specifically in the paragraph 3.1 (“Description of the ENDO-miRNA cohort”), the authors could describe the statistically significant differences in the demographic characteristics between the Control and Endometriosis groups. In the Table 1, the authors could replace the commas with dots and add the full form for rASRM classification, as they already did for BMI abbreviation. Moreover, the authors could check the mean and SD values for some characteristics, inserting only the SD values in the brackets; for example, Mean of Dyspareunia (±SD). In the paragraph 3.3 (“Feature selection of miRNAs relevant for a diagnosis of endometriosis”) the authors could check the symbols used for values higher or lower than 60% and 80, from 200 to 204 line.
  • In the Discussion section, at line 232, the authors could insert the full form for “AI”, since this word appears here for the first time. Finally, the authors could discuss more about the differences into the characteristics found between the two groups analyzed.
  • Minor points:

    -           In the abstract section, the authors could insert the full form for the abbreviation “AUC” and could explain better this parameter and the percentage assigned to the diagnostic miRNA signature.

Author Response

All comments of the reviewer have been answered. 

Again, the authors would like to sincerely thank the reviewer for the time spent in reviewing our work and for the comments that greatly improved this revised version.

Reviewer 2 Report

This is the first report linking salivary miRNA with endometriosis. If translated, this could be used as a diagnostic tool for endometriosis. This manuscript can be accepted in its current form. 

Author Response

Thank you for the time spent reviewing our work. We were pleased to read you found this work of high value.

Reviewer 3 Report

The authors have written a very interesting paper on a saliva-based miRNA signature for the diagnosis of endometriosis. The authors reported that an miRNA signature including 109 miRNAs had a very high sensitivity and specificity for differentiating endometriosis patients from controls. While this research is very important for advancing non-invasive diagnostic tools for endometriosis, there are several areas that could use revision in the manuscript, which have been separated into major and minor comments. Major comments 1. Toning down the conclusions of the paper – While the results of this study are impressive the conclusions that are drawn are too definitive. This study had a fairly small sample size, with less than 50 controls included. Additionally, there was no validation conducted of the miRNA signature on an external study population. Based on this the sentence in the Discussion on Lines 286-288 is an overstatement. While these results are promising, they do not justify a revision of the current diagnostic strategy and do not make diagnostic laparoscopy obsolete. It needs to be acknowledged that this study population was among women who were already pretty far along the pathway to an endometriosis diagnosis and so it is currently not known if this miRNA signature would perform well as an early marker of disease among adolescents and women who have not been suffering from endometriosis as long. Additionally, to better evaluate the study population for which this miRNA diagnostic signature may be applicable to, it would be useful to know the time from symptom onset to diagnosis and the proportion of participants who have infertility. Minor comments Abstract 1. Line 51 – what do you mean by “validated” saliva signature? Validated is often used to describe a measure that corresponds well with a gold standard. Your results have not been externally validated and so it is misleading and confusing to use the phrase “validated” saliva signature in the abstract. Introduction 2. Line 64 – It is not clear what “biological” means here and based on the references appears to mean biomarkers. It might be more informative to change this to “biological markers” or “biomarkers”. 3. Line 98-99 – What was inadequate about the control groups of previous studies? And how does this inadequacy compare to the control group used in this study? Methods 4. Lines 124-126 – What is meant by “importance of each symptom”? It appears to be the severity of their pain as this is what the Numerical Rating Scale measures but it is not clear from the methods or Table 1. 5. Lines 173-174 – Normally, without adjustment for multiple testing p-values of 0.05 or less are considered significant. As this study has many statistical tests, the p-value corrected for multiple testing should be much lower. 6. Line 178 – It would be useful for the reader to include a description of Random Forest so that the readers will have a better understanding of the methods. 7. Line 181-183 – More description about the use of the 10 data sets to test the accuracy and reproducibility of the signature is needed. Was the study population of 200 split among the 10 data sets? Was a test-retest design used? Table 1 8. It would be interesting to see the range of ages for the participants in the results or in Table 1. 9. It would be easier to understand the table if the N(%) was included for each row that shows a sample size and percentage instead of at the column headers. 10. Why were some of the pain symptoms presented as N(%) and some as a mean(SD)? And what is the mean(SD) of, the Numerical Rating Scale? There needs to be a footnote to explain what measure was used to calculate the mean and SD for the pain symptom variables. Was the Numerical Rating Scale used for the other pain symptoms, such as dysmenorrhea? As it is confusing right now with some pain symptoms presented as N(%) and some as mean(SD), I would suggest using one or the other. Alternatively, you could include both the N(%) and mean(SD) for all of the pain symptoms, if that information is available, as they are both informative about the study population. 11. For diagnostic method, the 47(100%) for the controls is in the middle of the rows. Did all of the controls have surgery or MRI? Results 12. Section 3.3 is difficult to read. Lines 227-233 is one very long sentence. It might be useful to show this in a figure or table so that the reader can more easily understand the paragraph. Additionally, it is not clear that the numbers in Lines 225-233 are about single miRNAs from the 109 selected by the algorithm. 13. Section 3.4 as mentioned in the methods, more description about these 10 datasets and how they were constructed, including their sample sizes, would be really useful in reading the paper. Additionally, Lines 239-241 mention the “best model”, did this model include all 109 miRNAs or were different combinations of miRNAs used across the 10 different data sets? Discussion 14. Lines 306-308 – while Moustafa et al are the only other ones to have used Random Forest, it is worth noting in the discussion that quite a few other studies have also looked at miRNA expression for the diagnosis of endometriosis. A lot of this paragraph is about findings in cancer and other conditions with very little space devoted to previous findings for endometriosis. References 15. The paper could benefit with the addition of references for some of the information presented Lines 58-59 Line 78 – 60% of genes regulated by miRNAs

Author Response

RESPONSE TO REVIEWER

Manuscript No:

Title: Salivary MicroRNA signature for Diagnosis of endometriosis

Corresponding Author: Dr. Sofiane Bendifallah

By Sofiane Bendifallah (MD, PhD) 1,2, Stéphane Suisse 3, Anne Puchar (MD) 1,2, Léa Delbos (MD) 4,5, Mathieu Poilblanc (MD) 6,7, Philippe Descamps (MD, PhD) 4,5, Francois Golfier (MD, PhD) 6,7, Ludmila Jornea (Msc) 8, Delphine Bouteiller (MD) 9, Cyril Touboul (MD, PhD) 1,2, Yohann Dabi (MD) 1,2 , Emile Daraï (MD, PhD) 1,2.

First, we would like to thank reviewer 3 for all its constructive and relevant comments.

All comments were considered carefully.

A detailed response has been formalized for each of them in the above document.

We hope that these substantial changes, considering their relevance and their clinical impact will find a favorable issue.

Reviewer 3

The authors have written a very interesting paper on a saliva-based miRNA signature for the diagnosis of endometriosis. The authors reported that an miRNA signature including 109 miRNAs had a very high sensitivity and specificity for differentiating endometriosis patients from controls. While this research is very important for advancing non-invasive diagnostic tools for endometriosis, there are several areas that could use revision in the manuscript, which have been separated into major and minor comments.

Thank you for this positive comment

Major comments 1.

  1. Toning down the conclusions of the paper – While the results of this study are impressive the conclusions that are drawn are too definitive. This study had a fairly small sample size, with less than 50 controls included.

We totally agree with this comment. Effectively, even our study is the largest available on miRNA on saliva (n=200), the sample size especially concerning the control group requires an external validation. Therefore, we suggested to add a sentence in the limit section:

“Although, our prospective study is the largest available on miRNA and saliva (n=200) [69], the sample size especially concerning the control group (n=47) and the internal cross validation imposes an external validation.”

As suggested, we rephrase the first sentence of the conclusion toning down our initial message.

“Despite some limits of the current prospective study, our data support the use of a saliva-based diagnostic miRNA signature for endometriosis in the diagnosis care pathways”.

  1. Additionally, there was no validation conducted of the miRNA signature on an external study population.

We totally agree with the comment. As aforementioned, an external validation is required to confirm this signature.

We suggested to add a sentence in the limit section:

“Although, our prospective study is the largest available on miRNA and saliva (n=200) [69], the sample size especially concerning the control group (n=47) and the internal cross validation imposes an external validation.”

  1. Based on this the sentence in the Discussion on Lines 286-288 is an overstatement. While these results are promising, they do not justify a revision of the current diagnostic strategy and do not make diagnostic laparoscopy obsolete.

We totally agree with the comment.

We suggest rephrasing the sentence as follow:

“These values testify to the high accuracy of the signature, supporting its clinical value, and justify a revision of the current diagnostic strategy for exploring patients with symptoms suggestive of endometriosis, rendering the gold standard of diagnostic laparoscopy obsolete.”

to

“These values testify to the high accuracy of the signature, supporting its clinical value, and raise the issue to revise the current diagnostic strategy for exploring patients with symptoms suggestive of endometriosis, based on a diagnostic laparoscopy.”

  1. It needs to be acknowledged that this study population was among women who were already pretty far along the pathway to an endometriosis diagnosis and so it is currently not known if this miRNA signature would perform well as an early marker of disease among adolescents and women who have not been suffering from endometriosis as long.

We totally agree with the comment concerning the specific issue of the endometriosis diagnosis for the adolescent population. As mentioned in the material and methods section only patients over 18 years old were included in the endomiRNA study. Even some patients were aged between 18 and 19 years old (according to the WHO definition, adolescent patients between 10 and 19 years old) only few patients included in the study can be considered adolescent. So, it is not possible to claim that our signature is relevant in the adolescent population.

Therefore, we suggest adding a sentence in the limit section:

“Our signature exhibits a high accuracy in patients over 18 years old. However, our population did not include enough adolescents. Therefore, it is not possible to extrapolate our results in this specific population imposing further studies.”

It is not possible to evaluate the relevance of our signature in asymptomatic patients. Indeed, all the patients included were symptomatic. Beyond this issue as no specific treatment is available for endometriosis and in accordance with national and international guidelines, it is not recommended to explore asymptomatic women with endometriosis. 

  1. Additionally, to better evaluate the study population for which this miRNA diagnostic signature may be applicable to, it would be useful to know the time from symptom onset to diagnosis and the proportion of participants who have infertility.

As suggested, we added data in the text the data on the delay on the symptom onset and diagnosis.

“The mean (± SD) time from symptom onset to diagnosis for endometriosis patients was 14.8 years (±17.88).”

Moreover, we added the number of patients with infertility in the control and endometriosis group in table 1.

Control

patients

N (%)

N=47

Patients with endometriosis

N (%)

N=153

Infertility

-        Yes

-        No 

17% (8)

83% (39)

24% (36)

76% (117)

0.556

Minor comments

  1. Abstract 1. Line 51 – what do you mean by “validated” saliva signature? Validated is often used to describe a measure that corresponds well with a gold standard. Your results have not been externally validated and so it is misleading and confusing to use the phrase “validated” saliva signature in the abstract.

In the statical analysis, we performed an internal cross validation according to 10 random datasets.

We propose to clarify the abstract by adding the following sentences:

“Of the 2 561 known miRNAs, the feature selection with Random Forest algorithm generated after internally cross validation a saliva signature of endometriosis composed of 109 miRNAs. The, respective sensitivity, specificity, and AUC for the diagnostic mi RNA signature were 96.7%, 100%, and 98.3%.

We propose to clarify the introduction by adding the following sentences:

“Therefore, the aim of the prospective ENDO-miRNA study was to analyze the human miRNome to differentiate patients with and without endometriosis, to define a saliva-based diagnostic miRNA signature for endometriosis with an internal cross validation.”

  1. Introduction 2. Line 64 – It is not clear what “biological” means here and based on the references appears to mean biomarkers. It might be more informative to change this to “biological markers” or “biomarkers”.

We agree with this comment.

We replace the following sentence

“The second challenge is that complementary explorations, especially biomarkers [5,6] and imaging examinations [7–10]”

  1. Line 98-99 – What was inadequate about the control groups of previous studies? And how does this inadequacy compare to the control group used in this study?

We considered that in previous studies the control groups was not adequate as included either asymptomatic patients, and/or patients requiring tubal ligation and/or patients with genital pelvic inflammatory disorder (PID). Even the study from Moustafa et al., all the control had others gynecological disorders without knowledge on symptom suggestive of endometriosis. In contrast, in the current study, all patients with or without gynecological disorders shared symptoms suggestive of endometriosis.

We add the following sentence in the introduction section  

“Indeed, in previous studies the control groups was composed of asymptomatic patients and/or patients undergoing a tubal ligation and/or pelvic inflammatory disease and/or gynecologic disorders but without knowledge on symptom [20,21,23,24].”

  1. Methods 4. Lines 124-126 – What is meant by “importance of each symptom”? It appears to be the severity of their pain as this is what the Numerical Rating Scale measures but it is not clear from the methods or Table 1.

We consider the comment.

As suggested we add in the manuscript the following sentence to clarify :

“Using Visual Analogical Scale (VAS), values of symptoms for patients with and without endometriosis were for dysmenorrhea (mean ± SD) 6 ± 3.4 versus 5 ± 3.2, p < 0.001, for pain on sexual intercourse were 3.8 ± 3.5 versus 2.3 ± 3.0, p < 0.001, and for urinary pain during menstruation 1.4 ± 2.5 versus 0.5 ± 1.4, p < 0.001, respectively.”

In table 1 we improve the quality of the reporting in accordance with comment 9.

  1. Lines 173-174 – Normally, without adjustment for multiple testing p-values of 0.05 or less are considered significant. As this study has many statistical tests, the p-value corrected for multiple testing should be much lower.

We used p-values of 0.05 do denote a statistical significance only for epidemiological characteristics reported in the table 1. We do not used any adjustment for multiple testing.

  1. Line 178 – It would be useful for the reader to include a description of Random Forest so that the readers will have a better understanding of the methods.

As suggested, we added additional data concerning the random forest method.

“Random Forest (RF) classifier is an ensemble method that trains several Decision Tree (DT) in parallel with bootstrapping followed by aggregation, jointly referred as bagging. Bootstrapping indicates that several individual DTs are trained in parallel on various subsets of a training dataset using different subsets of available features. Bootstrapping ensures that each individual DT in the RF is unique, which reduces the overall variance of the RF classifier. For the final decision, RF classifier aggregates the decisions of individual DTs and consequently exhibits good generalization [56].”

  1. Line 181-183 – More description about the use of the 10 data sets to test the accuracy and reproducibility of the signature is needed. Was the study population of 200 split among the 10 data sets? Was a test-retest design used? Table 1

The signature accuracy composed of 109 mi RNAs was internally validated on 10 different random datasets composed of the same proportion of control (n=47) and endometriosis (n=153) patients originate from the EndomiRNA study (n=200).

We added in the results section the following sentence

“The signature, after internal cross validation on 10 different data sets, obtain its higher accuracy with a respective sensitivity, specificity, and AUC were 96.7%, 100%, and 98.3% (Table 2).”

We clarify the abstract by adding the following sentences:

“Of the 2 561 known miRNAs, the feature selection with Random Forest algorithm generated after internal cross validation a saliva signature for diagnosis of 109 miRNAs. The, respective sensitivity, specificity, and AUC for the diagnostic mi RNA signature were 96.7%, 100%, and 98.3%.

We clarify the introduction by adding the following sentences:

“Therefore, the aim of the prospective ENDO-miRNA study was to analyze the human miRNome to differentiate patients with and without endometriosis, to define a saliva-based diagnostic miRNA signature for endometriosis with an internal cross validation.”

  1. It would be interesting to see the range of ages for the participants in the results or in Table 1. 9. It would be easier to understand the table if the N(%) was included for each row that shows a sample size and percentage instead of at the column headers.

As suggested, we added the range of age for the participants according to median in table 1.

Control

patients

N (%)

N=47

Patients with endometriosis

N (%)

N=153

Age (years) (mean ± SD)

30.92 (13.79)

31.17 (10.78)

0.1912

Age range

-        Less than 30 years

-        Over 30 years

72% (34)

28% (13)

63% (96)

37% (57)

0.294

  1. Why were some of the pain symptoms presented as N(%) and some as a mean(SD)? And what is the mean (SD) of, the Numerical Rating Scale? There needs to be a footnote to explain what measure was used to calculate the mean and SD for the pain symptom variables. Was the Numerical Rating Scale used for the other pain symptoms, such as dysmenorrhea? As it is confusing right now with some pain symptoms presented as N(%) and some as mean(SD), I would suggest using one or the other. Alternatively, you could include both the N(%) and mean(SD) for all of the pain symptoms, if that information is available, as they are both informative about the study population.

We consider the comment. We used number and percentage for qualitative data and mean and SD for quantitative data.

As suggested by the reviewer, we used only (n%) to describe the symptom in table 1.

Table 1.   Demographic Characteristics of the population

Control

patients

N (%)

N=47

Patients with endometriosis

N (%)

N=153

Age (years) (mean ± SD)

30.92 (13.79)

31.17 (10.78)

0.1912

Age range

-        Less than 30 years

-        Over 30 years

72%(34)

28%(13)

63% (96)

37% (57)

0.294

BMI (body mass index)  (mean ± SD)

24.84 (11.10)

24.36 (8.38)

0.525

rASRM classification 

-        I-II

-        III-IV

-

-

52%  (80)

48%  (73)

-

Infertility

-        Yes

-        No 

17% (8)

83% (39)

24%(36)

76%(117)

0.556

Control diagnoses

-        No abnormality

-        Leiomyoma

-        Cystadenoma

-        Teratoma

-        Other gynecologic disorders

51%  ( 24)

2%  (1)

11%  (5)

23%  (11)

13%  ( 6)

_

-

Dysmenorrhea

100%

100%

Abdominal pain outside menstruation

-              Yes

66%  (21)

71%  (89)

0.6905

Pain suggesting sciatica

-              Yes

31%  (10)

56%  (70)

0.0214

Lower back pain outside menstruation

-                Yes

62%  ( 20)

81%  (101)

0.0498

Right shoulder pain during menstruation

-                Yes

9%  ( 3)

21%  (26)

0.2184

Blood in the stools during menstruation

-                Yes

12%  ( 4)

24%  (30)

0.2425

Blood in urine during menstruation

-                Yes

25%  (  8)

17%  ( 21)

0.4172

Concerning pain symptoms, all the information were added in the main manuscript in the results section.

“Using Visual Analogical Scale (VAS), values of symptoms for patients with and without endometriosis were for dysmenorrhea (mean ± SD) 6 ± 3.4 versus 5 ± 3.2, p < 0.001, for pain on sexual intercourse were 3.8 ± 3.5 versus 2.3 ± 3.0, p < 0.001, and for urinary pain during menstruation 1.4 ± 2.5 versus 0.5 ± 1.4, p < 0.001, respectively.”

  1. For diagnostic method, the 47(100%) for the controls is in the middle of the rows. Did all of the controls have surgery or MRI?

As suggested by a previous reviewer and academic editor we added in the revised version the characteristics of the population including the control group

For the control groups (n=47), all the patients underwent a laparoscopy as diagnostic method with systematic histological biopsy.

For the patients in the endometriosis group without laparoscopic evaluation, all had MRI features of deep endometriosis with colorectal involvement and/or endometrioma which have been revised in multidisciplinary endometriosis committee.

To improve the quality of the manuscript due the importance of this issue, we propose to add the following information concerning the diagnostic method in table 1.

Diagnostic method

Control

Endometriosis

-

Surgery

47 (100)

83 (54.2)

-

Magnetic Resonance Imaging

-

70 (45.8)

-

  1. Results 12. Section 3.3 is difficult to read. Lines 227-233 is one very long sentence. It might be useful to show this in a figure or table so that the reader can more easily understand the paragraph.

As suggested, we simplified the results section 3

“Among the 109 miRNAs composing the endometriosis diagnostic signature, 77% (84) have been reported to be associated with pathophysiologic pathways for benign and malignant disorders (Annex 3). Only miR-34c-5p and miR-19b-1-5p have previously been reported in the field of endometriosis. Among the 109 mi RNA of the signature, 29 (27%) are associated with the main signaling pathways of endometriosis: PI3K/Akt, PTEN, Wnt/β-catenin, HIF1α/NF κB and YAP/TAZ / EGFR (annex 3).”

All additional information has been summarized in annex 3

Additionally, it is not clear that the numbers in Lines 225-233 are about single miRNAs from the 109 selected by the algorithm.

“The overall performance of the diagnostic signature composed of 109 mi RNAs (Random Forest model) against the 10 randomized datasets is reported in table 2. The sensitivity, specificity, and AUC ranges from 80% to 96.8%, 80% to 100%, and 79.9% to 98.4%, respectively. The signature, after internal cross validation on 10 different data sets, obtain its higher accuracy with a respective sensitivity, specificity, and AUC were 96.7%, 100%, and 98.3% (Table 2).”

  1. Section 3.4 as mentioned in the methods, more description about these 10 datasets and how they were constructed, including their sample sizes, would be really useful in reading the paper.

We consider this relevant comment.

To perform the validation accuracy of the signature we used 10 different data sets composed of endometriosis and control patients. Each data set was randomly generated to conserve the initial ratio of endometriosis and control patient’s profile.

We suggest adding a section in the material and method

Validation of the signature accuracy 

The accuracy and reproducibility of the signature were tested on 10 data sets randomly [61][62][63] composed of the same proportion of control, and endometriosis patients. Each data set was randomly generated to conserve the initial ratio of endometriosis and control patient’s profile.

We suggest adding a section in the limit:

“Although, our prospective study is the largest available on miRNA and saliva (n=200) [69], the sample size especially concerning the control group (n=47) and the internal cross validation imposes an external validation.”

  1. Additionally, Lines 239-241 mention the “best model”, did this model include all 109 miRNAs or were different combinations of miRNAs used across the 10 different data sets? Discussion

We consider this relevant comment.

We confirm that the signature included the combination of 109 mi RNAs selected.

This signature has been internally validated across 10 different data sets composed of control and endometriosis patients from the initial population.

We suggest rephrasing the sentence by :

“The signature, after internal cross validation on 10 different data sets, obtain its higher accuracy with a respective sensitivity, specificity, and AUC were 96.7%, 100%, and 98.3% (Table 2).”

We removed For the best model (n°4), representing

We suggested to add a sentence in the limit section:

“Although, our prospective study is the largest available on miRNA and saliva (n=200) [69], the sample size especially concerning the control group (n=47) and the internal cross validation imposes an external validation.”

  1. Lines 306-308 – while Moustafa et al are the only other ones to have used Random Forest, it is worth noting in the discussion that quite a few other studies have also looked at miRNA expression for the diagnosis of endometriosis. A lot of this paragraph is about findings in cancer and other conditions with very little space devoted to previous findings for endometriosis.

We take into account the comment and suggest adding the following sentence to discuss this issue in the manuscript.

“Several studies have reported aberrant expression of miRNAs in affected tissues or peripheral blood samples of patients with endometriosis [65-70]. Several miRNAs have been shown to be dysregulated during the pathogenic process of endometriosis [67] [66] [71]. Diagnostic power of several miRNAs has been assessed in endometriosis [67] [66] [71] [65] [70]. For example, Maged et al. have shown that serum miR-122 and miR-199a had a sensitivity of 95.6 and 100.0% and a specificity of 91.4 and 100%, respectively, for diagnosis of disease status in women. Thus, these miRNAs are putative serum biomarkers for endometriosis [72].”

  1. References 15. The paper could benefit with the addition of references for some of the information presented Lines 58-59 Line 78 – 60% of genes regulated by miRNAs

We understand the comment.

We added in the reference Lines 58-59 and Line 78 the following references.

  1. Bartel, D.P. MicroRNAs: Genomics, Biogenesis, Mechanism, and Function. Cell 2004, 116, 281–297, doi:10.1016/s0092-8674(04)00045-5.
  2. Hammond, S.M. RNAi, MicroRNAs, and Human Disease. Cancer Chemother. Pharmacol. 2006, 58 Suppl 1, s63-68, doi:10.1007/s00280-006-0318-2.
  3. Bartel, D.P. MicroRNAs: Genomics, Biogenesis, Mechanism, and Function. Cell 2004, 116, 281–297, doi:10.1016/s0092-8674(04)00045-5.
  4. Bartel, D.P. MicroRNAs: Target Recognition and Regulatory Functions. Cell 2009, 136, 215–233, doi:10.1016/j.cell.2009.01.002.

Round 2

Reviewer 3 Report

The authors have done a great job of incorporating all of the comments. I have only two minor suggestions for the revised manuscript.

1. Change "imposes" in Lines 337 and 340 to "warrants" or "requires".

2. I think it would be beneficial in the conclusion paragraph to restate that external validation is needed to confirm these results.

Author Response

RESPONSE TO REVIEWER

Manuscript No:

Title: Salivary MicroRNA signature for Diagnosis of endometriosis

Corresponding Author: Dr. Sofiane Bendifallah

By Sofiane Bendifallah (MD, PhD) 1,2, Stéphane Suisse 3, Anne Puchar (MD) 1,2, Léa Delbos (MD) 4,5, Mathieu Poilblanc (MD) 6,7, Philippe Descamps (MD, PhD) 4,5, Francois Golfier (MD, PhD) 6,7, Ludmila Jornea (Msc) 8, Delphine Bouteiller (MD) 9, Cyril Touboul (MD, PhD) 1,2, Yohann Dabi (MD) 1,2 , Emile Daraï (MD, PhD) 1,2.

We would like to thank reviewer 3 for all its constructive comments to improve the quality of the manuscript.

All comments were considered carefully.

A detailed response has been formalized for each of them in the above document.

We hope that these substantial changes, considering their relevance and their clinical impact will find a favorable issue.

Reviewer 3

The authors have done a great job of incorporating all of the comments. I have only two minor suggestions for the revised manuscript.

  1. Change "imposes" in Lines 337 and 340 to "warrants" or "requires".

We consider the comment and have made the changes as follow:

“Although, our prospective study is the largest available on miRNA and saliva (n=200) [76], the sample size especially concerning the control group (n=47) and the internal cross validation warrants an external validation.”

  1. I think it would be beneficial in the conclusion paragraph to restate that external validation is needed to confirm these results.

“Despite some limits of the current prospective study, our data support the use of a saliva-based diagnostic miRNA signature for endometriosis in the diagnosis care pathways after an external validation to confirm these results.